# Material Overconsumption as Ecological Polemics in Allen Ginsberg's "Plutonian Ode" and Gary Snyder's "Smokey the Bear Sutra": Re-Envisioning Beat Critiques of Anthropocentric Materialism

Henrikus Joko Yulianto

English Department, State University of Semarang, Semarang 50229, Indonesia; henriungaran@gmail.com

**Abstract:** Beat poetry, since its origination in the American milieu in the 1950s until its further maturation in the late 1960s and 1970s, has embodied ecological visions. Allen Ginsberg's and Gary Snyder's Buddhist poetics of the emptiness of material phenomena evoke one's awareness ofthe true nature of material goods. This ecological awarenessenlightensanyoneto not overconsume the goods in fulfilling his/her daily necessities. In this recent era, Ginsberg's "Plutonian Ode" and Snyder's "Smokey the Bear Sutra" memorialize this Beat green poetics against anthropocentric materialism and its potential detrimental impacts on the natural environment. These poems view human's material attachment as a recurring melancholia even in today's digital technology era. Their ecological criticisms through the Buddhist poetics pave the way for anyone to cherish rather than objectify any material thing in living the biotic community.

**Keywords:** Beat poetry; anthropocentric materialism; Buddhist poetics; biotic community

## 1. Introduction

Beat poetry has an ecological swing. It not only eulogizes jazz as its poetic form butserves as an ecological critique of mainstream material overconsumption. The notion of spontaneity that Beat writers, especially Jack Kerouac and Allen Ginsberg, learnt fromjazz rhythm and from Buddhist Mahāyāna teachings about the emptiness of phenomenahas entreated anyone to not objectify and attach oneself to material things (Allen, pp. 18–62; Yulianto 2017). The idea of 'spontaneity' in their poetics, as derived from bebop jazz and from the insight and manner Mahāyāna Buddhism evokes in anyone due to the insubstantiality of phenomena, serves as their non-materialist trajectory (Yulianto 2017). Indeed, Ginsberg's "Plutonian Ode" and Snyder's "Smokey the Bear Sutra" do not make use of bop prosody, which might be short and telegraphic, just as Jack Kerouac's choruses did in his *Mexico City Blues* in the era of the 1950s (Kerouac 1959). Instead, their long and asymmetric lines exemplify jazz improvization and emphasize urgency or 'immediacy' (Mortenson 2017, p. 81) as their spiritual poetics and critique of materialism. However, both Ginsberg and Snyder still embody a frenzied rhythm and the 'harsh, hard-edged' tonality of bebop in their use of enjambement that critiqued the material-oriented culture of people in their era (DeVeaux 1997). In fact, in "Plutonian Ode", Ginsberg still used interlocking strophes as he did in his 1950s poems such as "Howl" (Ginsberg 1956). Ginsberg's poems in the 1970s, such as "Plutonian Ode" in his *Plutonian Ode and Other Poems 1977–1980* (Ginsberg 1982), in a like manner raise environmental issues that reveal the era. "Plutonian Ode" is Ginsberg's prominent example of this poem with some ecological values. Its interlocking typography does suggest interconnectedness between things in the phenomenal world. Meanwhile, some years before, in 1969, the West Coast Beat poet Gary Snyder distributed his poem "Smokey the Bear Sutra" as a free broadside at the Sierra Club in San Francisco 'to promote ecological consciousness' (Charters 1992a, p. 519). Different from Ginsberg's ode, which

has a solemn tone and a musically rhythmic and regular pattern, Snyder's narrative and dramatic poem uses a sparse, irregular, and dadaistic typography. Synder expresses this latter feature through the use of capitalized lines and phrases that exemplify anger and protest. These capitalized lines are similar to those of his contemporary, Michael McClure, in his "Peyote Poem" written in 1958 (Charters 1992c, pp. 265–73). Then, the repeated phrases "DROWN THEIR BUTTS" and "CRUSH THEIR BUTTS" exemplify the poet's censure of the despoilers of the natural environment. Anthropocentric materialism is a never-ending phenomenon. From time to time, humans always depend on material things to fulfill their physiological and social needs. Any material products human beings invent will always serve their welfare. These human-oriented cultural activities exemplify the phenomenon of 'anthropocentrism' that initiated human hegemony overall life forms and the physical environment. In modern and contemporary lives, this anthropocentric activity has caused several threats tohuman and nonhuman animals as well as the natural environment.Ginsberg's long poemsfrom the era of the 1950s, such as "*Howl*", as well as those of the late 1950s, 1960s, and 1970s, such as "Kaddish", "Wichita Vortex Sutra", and "Plutonian Ode",portray the poet's bardic and elegiac laments about socio-political turmoils of these eras that all reveal the material anthropocentrism of those who held political supremacy (Katz 2016). In "Howl", Ginsberg more straightforwardly lampoons the social and ecological disharmony because of thisanthropocentric materialism. Inhis poem "For the Death of 100 Whales" that was read at the Six Gallery reading event in October 1955 in San Francisco (Morgan 2010, p. 102; Charters 1992b, p. 273), McClure censures the anthropogenic crime against whales that caused the demise of 100 whales. This marine mammal slaughter was certainly a crime against the endangered species that threatened biotic life in general (McClure 1982, p. 33). Then, McClure, as a West Coast poet, argued in an essay that the embodiment of human beings as being born of physical matter is derived from 'a spectrum of inorganic matter' and that we essentially coexist with other life forms (p. 118). He also argued that a man should be able to admit that he/she is an animal justlike anonhuman animal since all living beings similarly need biospheric elements, including oxygen, nitrogen, carbon, hydrogen, sulfur, and sunlight (pp. 115, 121). A female Beat poet of the younger generation, Anne Waldman, also recited a critique of human material overconsumption in her Buddhism-inspired poem "Makeup on Empty Spaces". For instance, some repeated lines in Waldman's poem, such as "painting the phenomenal world", "I bind the massive rock", and "I bind the uneconomical unrenewable energy of uranium" (Waldman 1989, pp. 130–31), all suggest a human craving for material things. This ecological awareness aims to make humans aware that they are not the center of life in the natural world but only parts of it just as other life forms are.

Human material overconsumption poses ecological hazards. This act will threaten the natural environment since it means to overexploit natural resources as the basic ingredients (Leonard and Conrad 2011). This material overextraction then will disorder the ecosystem. Mahayana Buddhist principles teach disciples to avoid material objectification because they posit the insubstantiality of any material thing (Simmer-Brown 2005; Capra 2005; Norberg-Hodge 2005; Gross 1997, pp. 291–311; Kaza 2010, pp. 39–61). Ginsberg in his poems "Ruhr-Gebiet" (1979) and "Homework" (1980) from the same anthology *Plutonian Ode & Other Poems 1977–1980* (Ginsberg 1982) critiques the human material overconsumption that has wreaked havoc on the natural environment and social harmony itself. For instance, in "Ruhr-Gebiet" he uses the phrases "too many" and "too much" to authenticate this material overconsumption—

Too much industry

No fish in the Rhine

Lorelei poisoned

Too much embarrassment

 … …

Too much metal

Too much fat

Too many jokes

not enough meditation

(Ginsberg 1982, p. 75)

This human act reveals his objectification of goods, which arises from his ignorance of the true essence of material things.

This article discusses Allen Ginsberg's "A Plutonian Ode" and Gary Snyder's "Smokey the Bear Sutra" as the two figuresare affiliated with the Beat Generation. The discussion is focused on human material overuseas ecological polemics. Two problems that this article discusses are: first, how Ginsberg's and Snyder's material poetics in the two poems reveal anthropocentric materialism and ecological polemics; and second, how their material poetics evoke one's ecological conscience in today's anthropocentric materialism. In discussing these questions, I refer to some books on Buddhist Ecology (Payne 2010; Tucker and Williams 1997; Badiner 2005), the nature of material objects in view of social anthropology (Bennett 2010; Miller 2005; Woodward 2007), and environmental humanities and material ecocriticism (Glotfelty and Fromm 1996; Iovino and Oppermann 2014). The reason for using these Buddhist ecological books is that the arguments in these books are based upon Buddhist teachings that evidently care about environmental issues and all life forms. Furthermore, Ginsberg and Snyder are two figures who, among other Beat poets, embraced Buddhism as their spiritual path and in their poetics (Yulianto 2017; Snyder 1969, pp. 90–93). Therefore, discussing their poems as ecopoems means to look into the Buddhist ecological views they embrace in their poems.

## 2. Anthropocentric Materialism as Human Phenomena

The term 'anthropocentric materialism' suggests human overuse of material things. These things are derived from natural resources and nonhuman organisms in the natural environment. The term 'materialism' is derived from the word 'matter', which means 'things', 'stuff', or often 'objects' (Miller 2005; Woodward 2007; Leonard and Conrad 2011; Bennett 2010). 'Matter' refers to any material phenomena in the natural world. This phenomenon certainly includes the ecology of all things or the 'mesh', the 'network', or the interconnectedness of things—humans, nonhuman animals, and material things in the natural environment. The mesh exists in all forms of life and material phenomena, including man-made products and natural resources such as plants, metals, fossil fuels, and any other material derivatives (Morton 2012, pp. 38–50). In the view of the environmental humanities and social anthropology, 'matter' exists in any biotic relationship, including the air one breathes and the food one consumes. This especially refers to material ecocriticism as a study that "examines matter both *in* texts and *as* a text" and aims to identify a human's interaction with his/her material things (Iovino and Oppermann 2014, pp. 1–3; Miller 2005). The idea of 'materialism' relates to consumerism that in the Buddhist perspective corresponds with human material overconsumption. This practice also tallies with overpopulation, while the latter phenomenon correlates with poverty (Gross 1997). Therefore, this excessive population and consumption of goods may suggest an anthropogenic propensity to crave for more material things. Furthermore, the American Buddhist writer and emeritus professor in Biology and environmental humanities Stephanie Kaza illustrated that materialism, which is identical with consumerism, in America has been around since the 1950s, the *era of the Beat Generation* (my emphasis). For instance, she said that since that year "the use of energy, meat, and lumber has doubled; use of plastic has increased five-fold; use of aluminum has increased sevenfold; airplane mileage has increased thirty-three-fold per person". In this era, people use cars twice as much as those living in the 1950s, such that they leave 'a larger ecological footprint' on the Earth (Simmer-Brown 2005, p. 3). Then, by referring to Loy's writing, Simmer-Brown said that consumerism may have become 'the new world religion', which is based upon the 'two unexamined beliefs' that "growth and enhanced world trade will benefit everyone" and "growth will not be constrained by the inherent limits of a finite planet" (Simmer-Brown 2005, pp. 3–4).

She further argued that the cause of consumerism (overconsumption of material goods) is 'ego gratification' or one's 'constant craving'. This human desire is derived from "the speed of one's minds, wishing so intensely for what one does not have that one cannot experience what is there, right before him/her" (Simmer-Brown 2005, p. 6). Referring to Buddhist teachings about the Noble Truth of Suffering (Goddard 1994), she pointed out that this consumerist craving causes suffering as she says "We want, therefore we consume; we want, therefore we suffer" (Simmer-Brown 2005, p. 4). Then, she referred to the notion proposed by Chögyam Trungpa Rinpoche, her *guru* and Ginsberg's Buddhist teacher, too, that materialism as the quality of one's reliance on materialness has three kinds—physical materialism, psychological materialism, and spiritual materialism (ibid., 2005, p. 5). Physical materialism means "the neurotic pursuit of pleasure, comfort, and security" or 'the outer expression of consumerism'. Psychological materialism refers to one's intention "to control the world through theory, ideology, and intellect". By having this ideology, one feels victorious, correct, and righteous. An example is the trend of contemporary people (laymen and public figures) to commercialize and objectify Buddhism for social popularity (Simmer-Brown 2005, p. 5). Spiritual materialism means one's attempt to use Buddhism to get rid of fear and insecurity and to maintain a 'centralized awareness' (Simmer-Brown 2005, p. 6). The history of materialism also closely corresponds with patriarchal culture, in which most people in modern times believe in the idea that 'real men' are those who possess more material wealth than others, especially women and children (Capra 2005, p. 11). In dealing with this materialism, the patriarchal culture connotes "expansion, competition, and an 'object-centered' consciousness" (Capra 2005, p. 11). Materialism in this millennium also corresponds with globalization (Norberg-Hodge 2005, p. 16). This global culture tends to produce homogeneity in cultural products, which is called a monoculture. This mass-culture not only means to expand and exploit material resources from the natural environment, but tends to privilege certain groups of industrialists to the disadvantage of villagers and working-class people. Since capital owners will replace farmers with 'capital-intensive machinery', diversified food production tends to take place with an export monoculture (Norberg-Hodge 2005, p. 17). In Ginsberg's "Plutonian Ode" and Snyder's "Smokey The Bear Sutra", the ecological polemics that the poems address reveal this patriarchal material-oriented culture. The use of uppercase letters in several phrases and images, such as 'Doctor Seaborg', 'Lord of Hades', 'the Great Year', 'Baptismal Word', 'Grand Subject', 'Solar System', 'Unapproachable Weight', and 'Diamond Arts', combined with the names of Greek and Hebrew gods and goddesses, such as 'Spring-green Persephone', 'Demeter', 'Sabaot', and 'Elohim' (Ginsberg 1982), in Ginsberg's poem suggest this patriarchal material-oriented culture (see Pederson 2009). Since most of the images written in uppercase letters are male or refer to male affairs and to what men conventionally do, these elucidate men's grasp on material things. In comparison, Snyder's use of uppercase letters in several phrases, sentences, and imperatives, such as 'SMOKEY THE BEAR', 'HE WILL PUT THEM OUT', and 'DROWN THEIR BUTTS' (Charters 1992a, pp. 570–71), depicts masculine superiority over any other gender, especially women. Since not only men but also women live in the natural world and they together constitute the agents who consume material things to survive, both men and women have to be aware of conserving biotic life. Therefore, Ginsberg's and Snyder's ecological vision in their poems evoke a male and a female reader's understanding of this urgency and responsibility.

Materialism always centers upon humans or *anthropos*. Everything that humans do is always for fulfilling their needs. Therefore, in daily material consumption, material goods and the natural environment become mere objects to cater to their necessities. The word 'anthropocentrism' then suggests 'a charge of human chauvinism' and 'an acknowledgement of human ontological boundaries (Boddice 2011, p. 1). In this millennium and global culture, anthropocentrism poses a polemic about what it means to be a human being (Sax 2011, p. 21), especially in dealing with other life forms and with human consumption of material goods. The word 'human' has an ecological overtone since it was derived from the Latin word *humanus* and this word originated from *humus*, which means 'earth' or 'soil'. Therefore, etymologically the term 'human' has already revealed the embodiment

of the earth in a human entity. This shows the inherent essence of the human ecology, the interconnectedness between humans and their natural environment. By the term 'anthropocentric materialism' I mean that human material overconsumption deals with one's greed. This anthropogenic activity means to disregard any other life forms, especially nonhuman animals. This human material overuse violates the inherent values of all life forms and material things when their entities play an important role in the biotic community (Meine 2013). Another meaning that I argue is that this anthropocentric materialism essentially depicts humankind's sense perception about material phenomena rather than the material things themselves as self-independent entities. Ginsberg's "Plutonian Ode" and Snyder's "Smokey The Bear Sutra" polemicize this psychological materialism through the illustration of environmental issues during the late 1960s and 1970s.

### 3. Ginsberg's and Snyder's Material Poetics as a Revelation of Anthropocentric Materialism and Ecological Polemics

Ginsberg's "Plutonian Ode" consists of three parts. Part I has 45 strophes; Part II has 12 strophes; and Part III has 8 strophes. So, this poem has 65 strophes in total. In a like manner, Snyder's poem more or less has 18 stanzas. The lengthiness of the poems in some ways suggests the materiality of natural phenomena and the human material overconsumption they criticize through the ironic and satirical overtones of 'Plutonian Ode' and 'Smokey The Bear Sutra'. Both poems use a lyrical, dramatic, and narrative style by the use of the first-person "I" and quoted speeches (with quotation marks). Their material poetics goes two ways. First, it ironizes and satirizes humankind's materialism through the polemics 'plutonium' and 'environmental despoliation'. Second, their material poetics is imbued with Mahāyāna Buddhist principles, namely the insubstantiality of the true essence of any material phenomenon. In Snyder's poem, he capitalizes some phrases and lines that highlight the issues. Capitalized phrases, such as SMOKEY THE BEAR, lines such as HE WILL PUT THEM OUT, "I DEDICATE MYSELF TO THE UNIVERSAL DIAMOND BE THIS RAGING FURY DESTROYED", (Charters 1992a, pp. 569–71) emphasize ecological and spiritual insights from the American west scene and eastern Buddhist teachings about the emptiness of phenomena (Snyder 1969, p. 92). Furthermore, the figure 'smokey the bear' emblematizes 'the US Forest Service campaign to control forest fires' (see Story of Smokey 2021; Wikipedia 2020c). The images 'the universal diamond' and 'highest perfect enlightenment' belong to the teachings of Mahāyāna Buddhism, which point toward emptiness as the true essence of phenomena. The image 'diamond' is derived from 'the Diamond Sūtra' or the 'Sūtra of the Diamond', a part of the Buddhist scriptures that teaches the highest wisdom of the true essence of phenomena as being devoid of self-entities. All material phenomena emerge in their interdependent relation to all other elements and to human consciousness (*pratītya-samutpāda*) (see Fischer-Schreiber et al. 2010, p. 57; Goddard 1994, pp. 85–107). The word 'diamond' in Sanskrit means *vajra* or *dorje* in Tibetan. In Buddhism, this suggests 'a symbol of the indestructible' ('adamantine') and stands for the emptiness (*shūnyatā*) of all material phenomena (Fischer-Schreiber et al. 2010, p. 241). This Buddhist principle of the interdependent arising is analogous with the idea of ecology, or the interconnection between living beings and the natural environment.

In a somewhat different manner, Ginsberg's poem uses an odic form, which might derive from a Greek ode. It is a long and meditative lyric poem with an elaborate stanza structure in various line lengths. Originally, an ode was a Greek choral song that was recited at religious festivals and described 'the adventures and sufferings of gods, goddesses, and heroes' (Hass 2017, pp. 209–10). This poem, like Ginsberg's other long poems, has indented and interlocking strophes that make a poetic ecology of the interconnection between one material thing and another. This poetic interconnectedness similarly reveals 'the interdependent arising' or *pratītya-samutpāda* of material phenomena. The title and subject of the poem, 'plutonian', clearly address ecological issues about the Earth's metallic material and its detrimental havoc when humans wrongly make use of it. Strophes 1–5 in Part I of Ginsberg's poem describe plutonium (uranium) as a material commodity that some American scientists experimented on as a nuclear weapon—

> What new element before us unborn in nature? Is there
> a new thing under the Sun?
> At last inquisitive Whitman a modern epic, detonative,
> Scientific theme
> First penned unmindful by Doctor Seaborg with poison-
> ous hand, named for Death's planet through the
> sea beyond Uranus
> whose chthonic ore fathers this magma-teared Lord of
> Hades, Sire of avenging Furies, billionaire Hell-
> King worshipped once
> with black-sheep throats cut, priest's face averted from
> underground mysteries in a single temple at Eleusis,
> (Ginsberg 1982, p. 11).

The 'new element' and 'chthonic ore' exemplify a material entity that humans crave to produce certain elements. The name "Dr. Seaborg" comes from the figure in history, Glenn Seaborg, an American nuclear chemist of Swedish descent who identified plutonium in 1944 (Bernstein 2007, pp. 74–77). It emerged from a neutron bombardment—neutron splitting or fission—from uranium-238, nucleus uranium-239, and neptunium-239 to, finally, plutonium-239 (Bernstein 2007, p. 76). Metallurgically, uranium is ancient and has been stored in the Earth's belly throughout the ages. The use of Greek and 'gnostic' gods and goddesses, such as Persephone, Demeter, Sabaot, Jehova, and Sophia, to describe this metal indicates the very ancient quality of the material as as well as anthropocentric materialism itself (see Pederson 2009). Then, the use of uppercase letters for plutonium-related images suggests patriarchal, material-oriented hegemony. This reflects Ginsberg's saying that, by sustaining this plutonium, the government wanted to set up 'a monolithic Surveillance State' (Schumacher 1992, p. 629). In fact, Ginsberg's poems always have political overtones but also articulate social and ecological views. Ginsberg's father, Louis Ginsberg, was the first son of Russian immigrants, Pincus and Rebecca Schectman Ginsberg. His mother, Naomi Livergant, was the daughter of a Jewish–Russian family, the Livergants. Louis taught English at some schools in New Jersey and wrote poems, while Naomi also taught at some schools there. Both the Ginsbergs and the Levys were individuals who were interested in liberal politics and social change. While the Ginsbergs were socialists, the Levys were devoted communists (Morgan 2006a, pp. 4–6). The family's craze for politics shaped Ginsberg's enthusiasm for political issues, too. The socio-political turmoil in the United States in the 1930s, or the Great Depression, might have been one cause of Naomi's mental illness. The continuous treatment of the mother in asylums not only distressed the family mentally but also expended much of the family's budget (Morgan 2006b, pp. 11–21). This family background might have contributed to the social–political nuances Ginsberg later embraced in his poems. For instance, in a letter to Richard Eberhart dated 18 May 1956, Ginsberg openly elucidated the features of his poem "Howl", which served as a form of social criticism. The goal of this criticism was "to offer constructive human values" and "to liberate basic human virtues" (Morgan 2008b, p. 137).Thus, while criticizing the governmental authority in the plutonium industry, his material poetics aimed to raise the public's ecological awareness of the hazards of the material when humans overconsume and mismanage it.

Next, scientists' discovery of transuranics (Bernstein 2007, pp. 63–65) reveals an ecological polemic since it is not the inherent quality of the metal that matters but the human cognizance of the atomic properties and of making them into a powerfully formidable agent.

The line "her daughter stored in salty caverns under white snow" (strophe 7) implies the poet's ecological polemical view that the metal should remain dormant in the Earth rather than be 'alive' due to being extracted by humans for their experiments. Another

example of this ecological polemic due to human perception rather than material perceptibility is the poet's confession about his supremacy in identifying the very properties of the material, such as in "I manifest your Baptismal Word after four billion years" and "I enter your secret places with my mind, I speak with your presence, I roar your Lion Roar with mortal mouth" (strophe 14 and 25) (Ginsberg 1982, pp. 12–13). The images in uppercase letters, such as 'Baptismal Word' and 'Lion Roar', do not clearly objectify plutonium as a self-existent material but essentially reveal the human mind in recognizing the formidable material.

Names such as Hanford, Pantex, and Washington (strophe 20) exemplify the nuclear reactors that scientists built during that time (Bernstein 2007). The phrase "a new Thing under the sun" (strophe 22) indicates the neutron fission that resulted in plutonium-239 and that this transuranics is not naturally made but is an anthropogenic experimentation. This likewise proves how this human-material-oriented mind, through the reactors, would have threatened all life forms since even a speck of uranium dust would harm living beings. Allen Ginsberg and his friends, for instance, protested against them by sitting on the rail track outside of Rockwell Corporation's Nuclear Plant in Colorado to demand the disarmament of nuclear production (Schumacher 1992, pp. 628–29). The names of several places, such as Hanger-Silas Mason and Manzano Mountain, where uranium processing and mining occur indicate this anthropocentric materialism (Wikipedia 2020a, 2020b). Strophe 24 illustrates the poet's alarm at the hazards of nuclear reactors through their toxic emissions. This is human recognition of transuranics and their potential power for armament that everyone should be scared of. In strophes 25 to 29, the poet psychically delves into the inner elements of plutonium and aims to cognize the true nature of the metal. In Buddhist practices, this way of interiorizing a material object to understand its nature is a form of mindfulness to 'wash away the toxins in one's psyche or consciousness' (Hanh 2005, p. 237)—"I enter your secret places with my mind, I speak with your presence, I roar your Lion Roar with mortal mouth" (strophe 25). This line also suggests the poet's use of a Buddhist and gnostic mantra as a means of recognizing one's attachment to the material as the reason rather than the plutonium as a self-existent materiality (see Pederson 2009, p. 34).

This strophe indicates a human way of 'othering' the plutonium without objectifying it because of its fearful qualities. The following strophes illustrate how hazardous even a speck of plutonium dust is to human and nonhuman beings and to the natural environment. At the same time, the poet's description of the frightful nature of the plutonium discloses the ecological fact that human beings and natural phenomena, including plutonium as one of the natural elements, are interconnected.

The phrase 'the Wheel of Mind' followed by 'your three hundred tons' (strophe 36) exemplifies this interconnectedness between one's consciousness and the plutonium. This means that the human consciousness has embedded the materiality of objects despite the fact that the objects have their own entities in a relative sense (Chandrakirti 2002, pp. 199–200). The next line, "I embody your ultimate powers", then emphasizes the superior agency of human consciousness in recognizing material phenomena and their materialities. The next line, "I sing your form at last", and the next, cataloguing "behind your concrete & iron walls inside your fortress of rubber & translucent silicon shields in filtered cabinets and baths of lathe oil" (strophes 37–38), further the poet's mindful mastery over the nuclear reactors.

In comparison, in Snyder's "Smokey The Bear Sutra", in stanza 1 the poet briefly analogizes the mythic account of Siddhartha Gautama, the founder of Buddhism, with the ecological interdependence between a living being and the natural elements and ecological hazards in the natural world by describing anthropogenic threats to nonhuman animals and the natural environment in general—

Once in the Jurassic about 150 million years ago,

the Great Sun Buddha in the corner of the Infinite

Void gave a Discourse to all the assembled elements



and energies: to the standing beings, the walking beings,

the flying beings, and the sitting beings—even grasses,

to the number of thirteen billion, each one born from a

seed, assembled there: a Discourse concerning

Enlightenment on the planet Earth.

(Charters 1992a, p. 569)

Some phrases and words written in uppercase letters, such as 'the Great Sun Buddha', 'the Infinite Void', 'Discourse', and 'Enlightenment', disclose the material aspects of these things. These capital letters also give prominence to these things as components of ecological life in the biosphere. Furthermore, the capital letters in the phrases 'the Infinite Void' and 'Enlightenment on the planet Earth' and in other words, especially 'the Sun Buddha' and 'Discourse', simultaneously aim to de-materialize these things that, despite being material things that emerge as 'material' things, are devoid of their true nature (Rinpoche and Larampa 2012, pp. 39–42). Then, 'the standing beings', 'the walking beings', 'the flying beings', and 'the sitting beings' suggest human and nonhuman animals that originally came from 'seed', where 'seed' itself signifies the origin of a life form. In Buddhism, *seed* itself does not essentially possess its own true existence but constitutes an aggregate of elements (Chandrakirti 2002, pp. 210–14). The word 'enlightenment' (*bodhi* in Sanskrit, *satori* in Japanese) or 'awakening' describes a state of one's awareness of the insubstantiality/emptiness of external phenomena (Fischer-Schreiber et al. 2010, p. 65). The poet's mentioning of 'the standing beings', 'the walking beings', 'the flying beings', 'the sitting beings', and 'grasses' exemplifies his warning to people in general to care for all life forms. The word 'enlightenment', as the true essence of phenomena, then subjugates one's material-oriented desires and his objectification of nonhuman animals and things. This word also evokes one's awareness of the interdependence between living beings in the natural environment. This ecological consciousness will encourage everyone to cherish all of the life forms that coexist as a biotic community (Meine 2013, pp. 171–89) or what Gary Snyder called *saṅgha* (Barnhill 1997, pp. 187–217). For instance, in stanza 6, the poet describes 'a handsome smokey-colored brown bear' as the character who aims to bring enlightenment to living beings—

Bearing in his right paw the Shovel that digs to the

truth beneath appearances; cuts the roots of useless attach-

ments, and flings damp sand on the fires of greed and war;

His left paw in the Mudra of Comradely Display—indicating

that all creatures have the full right to live to their limits

and that deer, rabbits, chipmunks, snakes, dandelions,

a nd lizards all grow in the realm of the Dharma;

(Charters 1992a, pp. 569–70)

While showing the poet's concern for endangered species like bears in the United States, the figure 'the smokey bear' serves as the natural emblem of the U.S. Forest Service campaign (Wikipedia 2020c). The spiritual overtone of the smokey bear likewise suggests the teaching of Mahāyāna Buddhism about the emptiness of material phenomena. The phrase 'the Mudra of Comradely Display' refers to the Buddhist word *mudrā*, which means 'a bodily posture or a symbolic gesture' (Fischer-Schreiber et al. 2010, p. 148). These gestures correlate to natural gestures (of teaching, protecting, etc.) and to certain aspects of Buddhist teaching. There is an ecological aspect since these gestures connect the practitioner with 'the buddha visualized in a given practice' (*sādhana*). There are 10 gestures: *dhyāni* mudrā (gesture of meditation), *vitarka* mudrā (teaching gesture), *dharmachakra* mudrā (gesture of turning the wheel of the teaching), *bhūmi-sparsha* mudrā (gesture of touching the earth), *abhaya* mudrā (gesture of fearlessness and granting protection), *varada* mudrā (gesture of granting wishes), *uttara-bodhi* mudrā (gesture of supreme enlightenment), mudrā of

supreme wisdom, *añjali* mudrā (gesture of greeting and veneration),and*vajrapradama* mudrā (gesture of unshakable confidence) (Fischer-Schreiber et al. 2010, p. 148). All of these mudrā likewise embody ecological aspects since they connect humans with each other, with nonhuman animals, and with the physical environment. Then, the image 'smokey the bear' itself refers to the actual bear that a Native American languagecalled the Large Brown One and the Old Man in the Fur Coat (Wikipedia 2020c). The poster of the smokey bear in wikipedia shows the bear's finger pointing to the reader, signifying *vajrapradama* mudrā or the 'gesture of unshakable confidence' since the pointing commandingly solicits everyone to keep forests from fires. At the same time, Snyder compares the bear with some Buddhist figures, including the Ancient Buddha, and some Buddhist teachers whom Snyder met when he lived and studied Buddhism in Japan in the 1950s and the 1960s (Snyder 1999, pp. 243–44; Yampolsky 1991, pp. 60–69). As the guardian of wildfires, the fire the bear quells is analogous with 'the fires of greed and war' that Siddhārtha Gautama aims to extinguish in humans (Snyder 1999, p. 244). Furthermore, in stanzas 9 and 12, the poet polemicizes several environmental issues that the West Coast areas were facing, including forest fires, manufactured canned foods, and an excessive number of vehicles—

> With a halo of smoke and flame behind, the forest fires
>
> of the kali-yuga, fires caused by the stupidity of those
>
> who think things can be gained and lost whereas in truth all
>
> is contained vast and free in the Blue Sky and Green Earth
>
> of One Mind;
>
> … …
>
> Indicating the Task: his followers, becoming free of cars,
>
> houses, canned foods, universities, and shoes, master the
>
> Three Mysteries of their own Body, Speech, and Mind; and
>
> fearlessly chop down the rotten trees and prune out the
>
> sick limbs of this country America and then burn the leftover
>
> trash.
>
> (Charters 1992a, p. 570)

The long line "fires caused by the stupidity of those who think things can be gained and lost whereas in truth all is contained vast and free in the Blue Sky and Green Earth of One Mind" exemplifies the human craving for material things. Then, the images 'vast and free', 'the Blue Sky', and 'Green Earth of One Mind' exemplify the materiality of the natural landscapes. At the same time, the capital letters in these natural images aim to de-materialize them in order that one will not regard them as having a true existence (Chandrakirti 2002, pp. 166–67). 'The Blue Sky' and 'Green Earth of One Mind', written in uppercase letters, are images used in Buddhism that mean 'a panoramic awareness' or 'a state of one's consciousness where there is no division between subject and object' and an awareness of the interdependence of material things on causes and conditions (Tonkinson 1995, p. vii). In the next stanza, the poet mentions some material things that human beings should stay away from, including excessive numbers of vehicles and canned foods. This overuse of vehicles and canned foods exemplifies anthropocentric materialism and causes environmental problems. For instance, the overuse of vehicles causes air pollution and canned food overconsumption will endanger the body's wellness. In 1969, Snyder wrote an essay entitled "Four Changes", in which one of the essays is about 'pollution'. The sources of pollution do not only come from chemical substances, such as DDT, but also from the fossil fuel combustion of the increasing number of vehicles (Snyder 1969, pp. 94–95). The image 'the Three Mysteries of Body, Speech, and Mind' is derived from a Buddhist teaching (dharma) of Vajrayāna that consists of 'specific bodily postures and gestures' (mudrā), concentration of the mind (samādhi), and the recitation of sacred syllables (mantra)'. Buddhism symbolizes this threefold aspect in 'many ritual

texts' by 'the seed syllables *om ah hum*. The syllable *om* in white in 'the forehead' refers to body'; the syllable *ah* in red to 'the throat center and speech'; and the syllable *hum* in blue to 'the mind' (Fischer-Schreiber et al. 2010, pp. 25–26). The term 'Vajrayāna' itself comes from the Sanskrit word *vajra*, which means 'diamond' or 'adamantine', while the Sanskrit word *yāna* points to 'vehicle'. So, the word in tandem means 'Diamond Vehicle' or a school of Mahāyāna Buddhism that emphasizes teachings about the emptiness of external phenomena (Fischer-Schreiber et al. 2010, pp. 241–42, 251). In portraying subjects imbued with Buddhist nuances, both Ginsberg and Snyder were kind of othering but also anthropomorphizing 'plutonium' and 'the smokey bear' to emphasize their ecological significance. They used parody or a 'camp aesthetic', which means "to diminish the value of the religious experience" and "to transmit their knowledge of Buddhism" to readers in generalwho are not familiar with Buddhism (Belletto 2017, p. 19; Whalen-Bridge 2017, p. 232).

## 4. Material Poetics as an Ecological Vision in the Era of the Anthropocene

Humankind's relation to materialism is an ecological network. It is what Buddhism calls 'the Indra's Net' or 'food-web community' (Barnhill 1997, pp. 188–91). Here, I also argue that material poetics in Ginsberg's and Snyder's polemics on environmental issues exemplifies 'the Indra's Net' because these issues reveal the interdependence of things as a virtue of ecological interconnectedness in the natural world. These polemics serve as a criticism of the human objectification of material goods. This criticism then serves as an agent to raise readers' ecological awareness because of the fact that material over-consumption has depleted natural resources. Furthermore, the hazards of plutonium and other related subterranean resources endanger all life forms in the physical environment. Ginsberg expresses these material poetics through his spiritual interiorization of plutonium as the epitome of ecological interconnectedness between human beings and a material object. Yet, his spiritual immersion through the Buddhist threefold division of body, speech, and mind raises one's awareness of the true nature of the object. Strophe 43 says: "Poured on the stone black floor, these syllables are barley groats I scatter on the Reactor's core", in which the image 'these syllables' represents speech as his skillful means (or called upāya in Mahāyāna Buddhism-see Fischer-Schreiber et al. 2010, p. 239) of controlling the superiority of the object. These syllables similarly serve as a mantra that aims to subdue one's material-oriented mind, especially his craving for material objects and their versatile materialities. He satirizes human efforts in inventing plutonium since scientists make use of this for armaments rather than for social welfare. Strophes 46 to 48 in Part II exemplify this human material attachment through the experiment of discovering plutonium—

> The Bard surveys Plutonian history from midnight
>
> Lit with Mercury Vapor streetlamps till in dawn's
>
> Early light
>
> He contemplates a tranquil politic spaced out between
>
> Nations' thought-forms proliferating bureaucratic
>
> & horrific arm'd, Satanic industries projected sudden
>
> With Five Hundred Billion Dollar Strength
>
> (Ginsberg 1982, p. 15)

At the same time, 'plutonium' also serves as a 'Beat' metaphor for a kind of 'bedrock consciousness' as the stripped off-mind the Beats experienced with the essence of things in the material world as John Clellon Holmes said in his article "This Is The Beat Generation" published in *New York Times Magazine* on 16 November 1952 (see McDarrah 1985, p. 22). The word 'bard' means 'poet' and so it refers to the Beats themselves. However, this also satirizes the scientists who discovered plutonium and made use of it for the armament industries. He expresses this critique through the image 'bureaucratic & horrific arm'd, Satanic industries projected sudden with five hundred billion dollar strength', which exem-



plifies human material-oriented desire. Furthermore, in strophes 60 to 65, Ginsberg once again chants his mantra by spiritually interiorizing plutonium and its formidable qualities as one's mind perceives them. This aims to identify the insubstantiality of plutonium as a self-autonomous element (and other metals and material things—my emphasis), so that human beings will not be excessively attached to it—

> Take this wheel of syllables in hand, these vowels and
>
> consonants to breath's end
>
> take this inhalation of black poison to your heart, breathe
>
> out this blessing from your breast on our creation
>
> forests cities oceans deserts rocky flats and mountains
>
> in the Ten Directions pacify with this exhalation,
>
> enrich this Plutonian Ode to explode its empty thunder
>
> through earthen thought-worlds
>
> Magnetize this howl with heartless compassion, destroy
>
> this mountain of Plutonium with ordinary mind
>
> and body speech,
>
> thus empower this Mind-guard spirit gone out, gone
>
> out, gone beyond, gone beyond me, Wake space,
>
> so Ah!
>
> (Ginsberg 1982, pp. 16–17)

Some extreme lines that imperatively encourage his readers to spiritually interiorize this plutonium, such as those in strophes 61 and 63, exemplify his critique of human indifference to ecological robustness and of his worldly material craving ('earthen thought-worlds'). The image 'the Ten Directions' in strophe 62 refers to one of the Buddhist principles (dharma) (Goddard 1994, pp. 653–55). These strophes as well as strophes 64 and 65 tend to separate the plutonium from human consciousness as if the metal element exists as a self-autonomous entity. Yet, the poet's images 'heartless compassion', 'ordinary mind and body speech', and 'gone beyond me' signify Buddhist teachings. The phrase 'ordinary mind and body speech' suggests the 'intuitive mind' rather than the 'discriminating mind' for an individual in viewing phenomena in this material world (Goddard 1994, pp. 306–7). The phrase 'gone beyond me' and mantra 'ah' likewise imply the insubstantiality of phenomena. This mantra simultaneously serves as the poet's skillful means of preventing the American government from enlarging the nuclear industry due to its environmental hazards (see Pederson 2009, pp. 34–35). Even more so, the United States and other large countries around the world still use uranium as the material to produce electrical energy and for other purposes in nuclear plants (see Cravens 2007).

In the last stanza of his poem, Snyder suggests the notion of the insubstantiality of material phenomena through the figure 'smokey the bear', who also recites a mantra to all living beings to discern this essence—

> And if anyone is threatened by advertising, air pollution, television,
>
> or the police, they should chant SMOKEY THE BEAR'S WAR SPELL:
>
> DROWN THEIR BUTTS
>
> CRUSH THEIR BUTTS
>
> DROWN THEIR BUTTS
>
> CRUSH THEIR BUTTS
>
> And SMOKEY THE BEAR will surely appear to put the enemy out
>
> with his vajra-shovel.
>
> (Charters 1992a, p. 571)

The first line above satirizes humankind's excessive consumerism and materialism. This satire similarly serves as an agent to raise the reader's ecological awareness since the lines above polemicize the material products that jeopardize humankind's life and counter these with the figure of smokey the bear as an endangered animal combined with a Buddhist spiritual mantra. The satirical mantra then raises the reader's awareness of the hazards that human material overconsumption will pose to the natural environment. The Smokey Bear's repeated mantra in capital letters substantiates the poet's concern about environmental problems caused by anthropogenic activities. The image 'butts' has various meanings. However, as a form of satire, it means 'a person's buttocks or asshole', a derogatory word for human indifference to the environmental issues. Then, in the last stanza, the poet points out some secure conditions for a person if he/she cares about the natural environment and recites the Sutra (Goddard 1994)—

> Now those who recite this Sutra and then try to put it in
>
> practice will accumulate merit as countless as the sands
>
> of Arizona and Nevada.
>
> Will help save the planet Earth from total oil slick.
>
> Will enter the age of harmony of man and nature.
>
> Will win the tender love and caresses of men, women, and
>
> beasts
>
> Will always have ripe blackberries to eat and a sunny spot
>
> under a pine tree to sit at.
>
> AND IN THE END WILL WIN HIGHEST PERFECT
>
> ENLIGHTENMENT.
>
> (Charters 1992a, p. 571).

The image 'countless sands of Arizona and Nevada' describes how a person will reap a spiritual enrichment as much as the sand in his/her insight into the true nature of phenomena. The subsequent lines using 'will' exemplify the 'conditioned arising', suggesting that a person will experience spiritual enlightenment if he/she is willing to liberate himself/herself from material obsession. The interdependent causality that the stanza reveals is called *pratītya-samutpāda* (conditioned arising) in Mahāyāna Buddhism (Fischer-Schreiber et al. 2010, p. 172). This means that spiritually enlightened figures or those who practise Buddhist wisdom in their daily life will undeniably cherish the natural environment and the life forms that persist in the natural world. Both Ginsberg and Snyder embrace the Buddhist teaching of compassion (*karunā*) toward all life forms. This spiritual insight cherishes the inherent values in all life forms, including both animate and inanimate beings. In Buddhism, these values are called Buddha Nature (Abe 1989, pp. 46–47). The idea is that each material phenomenon in this natural world coexists with human consciousness in perceiving it so that each thing has mind—"grasses, trees, and lands are mind; being mind, they are *shujō*; being *shujō*, they the Buddha-nature" (Abe 1989, p. 46). The insight into the inherent values of a material thing will evoke everyone to care rather than to waste the thing in his or her daily material consumption. The last line in capital letters emphasizes the goal of this spiritual quest, which is to attain enlightenment. In Buddhism, this state means the liberation of individuals from bondage with material attachment and also the insight into the emptiness of the true existence of material things.

Both Ginsberg and Snyder as American poets internalize Buddhist teachings in their poems. Buddhist nuances in their poems were not just persona, as they observed Buddhist practices, such as meditation and chanting sutra, until the era of the 1970s when Ginsberg wrote "Plutonian Ode" and Snyder wrote "Smokey the Bear Sutra" (Morgan 2008a, pp. 387–89; Snyder 1999, pp. 243–44). Their embrace of Buddhism came from their inward willingness rather than the collective drive of orientalism or American exceptionalism.

Indeed, the term 'orientalism' in a general sense can mean someone's activity in teaching, writing about, or researching things of the Orient (Said 1979, p. 2). In this case, Buddhism came from India and then developed in Japan and Tibet, which constitute the oriental regions. In some ways, Ginsberg and Snyder's observance of Buddhism can represent orientalism in a conventional sense without any stereotypical colonial desire. In a like manner, American exceptionalism and American innocence are socio-political attitudes that correlate with colonialism, capitalism, and white supremacy (Baraka 2019, p. 14). Their poetics certainly serve asa form of 'social rebellion' and they expressed this social criticism via their own voices as Beat poets who had learned Buddhism and its teachings rather than being suggestive of American exceptionalism or orientalism. With their Beat Buddhist poetics, they had more freedom to search for enlightenment for America in particular and to each individual reader in general.

## 5. Conclusions

Both Ginsberg and Snyder polemicize several environmental issues related to human material overconsumption. Apart from any other social and political overtones, their depiction of 'plutonium' and 'smokey the bear' in their poems conveys ecological insight. Their ardent voice and satirical tone in conversing about the subjects in a poetically chronological sequence elucidate their concern about the havoc that human overconsumption of plutonium and other material products may wreak on all life forms and the natural environment.Nowadays, the United States uses uranium as a biotic commodity and a mineral to produce electricity and energy in reactors (www.world-nuclear.org (accessed on 28 March 2021)). In comparison, Snyder's portrayal of forest fires and material products that humans overconsume daily, including vehicles, fossil fuels, and other commodity products, corresponds with modern social life in a global world in which consumerism and materialism are humankind's way of life. Even more so, the controversial issue of global warming and climate change will always haunt each country in the world that overconsumes materials such as fossil fuels that greatly contribute to a larger ecological footprint. Deforestation and land clearing for the plantation and mining industries still exist in several third-world countries (see Seymour and Busch 2016). These anthropogenic activities certainly devastate physical landscapes, emit carbon dioxide and toxic waste into the air and surrounding areas, and also deplete biodiversity in these areas. Ginsberg and Snyder's material poetics then serve as what Jack Kerouac once called 'a new vision' to raise one's moral and ecological awareness to not extravagantly pursue and consume material things. The ecological insight they convey is an adamantine poetics since it originates from Buddhist wisdom about the insubstantiality of material phenomena. In a conventional sense, this wisdom teaches everyone to cherish all life forms and to consume material goods sufficiently so that their acts will produce ecological resilience and sustainability.

**Funding:** This research received no external funding.

**Conflicts of Interest:** The author declares no conflict of interest.

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
