# Peer review of "Material Overconsumption as Ecological Polemics in Allen Ginsberg’s “Plutonian Ode” and Gary Snyder’s “Smokey the Bear Sutra”: Re-Envisioning Beat Critiques of Anthropocentric Materialism"

_humanities, doi:10.3390/h10020078_

Round 1

Reviewer 1 Report

This paper is intriguing, but will need copy editing to make the English more idiomatic.  The argument seems to be cohesive and supportable, and the paper gives clear insight into Buddhist content of the poems by Ginsberg and Snyder.  However, there are areas that could use tightening.

Plutonium, as I understand it, is an element separate from uranium based on its atomic weight.  On earth it has been created from uranium, and it is impossible to say if it exists somewhere independently in nature.  Of course, from a Buddhist perspective, the concept of elements itself might be considered anthropogenic materialness.

The word "materialness" is introduced early in the article, but only defined later.  It might be good to define it on first usage or to explain that it will be defined more fully later.  Is this a specific usage?  If so, where does it derive from?  How does "materialness" differ from the more common "materiality"?

Discussion of the origin of Smokey the Bear is confused.  "Large Brown one" and other characterizations of bear sound to me like Native American characterizations of bear, and do not refer to a person Snyder met.  Snyder probably met a bear!  See Snyder's rendering of the tale "The Girl Who Married a Bear" in his book Practice of the Wild.  The examples sound like Native metaphoric or personifying representations of bear as an entity.

Smokey the Bear is also a real bear who became emblematic in the US Forest Service campaign to control forest fires.  See the links below for details.

https://smokeybear.com/en/smokeys-history/story-of-smokey

https://www.smokeybear.com/en

https://en.wikipedia.org/wiki/Smokey_Bear

In this context "butts" are cigarette butts (discarded cigarettes being a major cause of forest fires) as well as “human” butts.  This is part of the satire.

The satire in both Ginsberg's and Snyder's poems could be emphasized.  This would fit in with the rough humor expressed in a saying like "If you see the Buddha, kill him."   See John Whalen-Bridge's chapter “Buddhism and the Beats,” in The Cambridge Companion to the Beats, edited by Steven Belletto (Cambridge University Press, 2017).  Whalen-Bridge suggests that the Beat immersion into Buddhist beliefs and practices is part of a long, ongoing introduction of Buddhism into American culture, starting in the early to mid nineteenth century.  Much of the effort gets absorbed into moments of popularization that tend to obscure the serious nature of the enterprise.  Whalen-Bridge points to the “camp aesthetic” that underlies much of the Beat attempt to transmit Buddhist beliefs.  Camp is defined as “a sympathetic form of parody” reflecting Susan Sontag’s description of comedic camp as “an experience of underinvolvement, of detachment” (qtd in Whalen-Bridge 232). Thus much of the working of Buddhism into Beat literature—Whalen-Bridge cites in particular Ginsberg’s “Wichita Vortex Sutra” and Snyder’s “Smokey the Bear Sutra”—involves an underplaying or semi-satiric commingling of enlightenment with entertainment, as in Snyder’s playful but austere assertion that “Smokey the Bear will illuminate those who would help him; but for those who would hinder or slander him, HE WILL PUT THEM OUT” (qtd in Whalen-Bridge 234).

Smokey the Bear points directly at the reader or viewer in posters that feature him.  Is this a mudra?  

I encourage revision and particularly attention to style and usage to make the work more idiomatic.

.  

Author Response

Thank you very much for your feedback and suggestions. Yes, I think that I have given a brief definition about 'materialness' in the introductory part of my article. I prefer using 'materialness' because this word refers to material things themselves. In comparison, the word 'materiality' means 'the quality of being material or composed of matter' that describes the contents of any material thing. Please feel free to give me more suggestion for a more exact term. 

Yes, I have checked the brief article in The Gary Snyder Reader about the figures whom he associated with 'the smokey bear'. I have also checked the links that you have suggested and inserted them in referencing. I do very apologize if I have missed some sources that you have suggested. It is because I was in a hurry to do the revision to my article and to send it back soon.

Yes, I have read the article by John Whalen-Bridge and inserted it as a reference to my article. Yes, I agree with the term that Whalen-Bridge used "camp aesthetic" to describe a humorous and playful manner Ginsberg and Snyder use to illustrate Buddhist nuances in their poems to general readers. Yes, all right. I also agree if the poster of the smokey bear pointing his finger at Wikipedia represents mudra, which means to raise each individual's understanding and  care about conserving forests in his/her areas.

Yes, I've checked and revised some sentences and deleted wordiness. Thank you very much.

Reviewer 2 Report

Biotic Commodities as Ecological Polemics in Allen Ginsberg’s “Plutonian Ode” and Gary Snyder’s “Smokey The Bear Sutra”: Re-Envisioning Beat Critiques of Anthropocentric Materialness 

General comments

This article addresses anthropogenic materialism and commodity and extractives cultures in the works of two Beat-generation poets: Gary Snyder and Allen Ginsberg. The article foregrounds some interesting arguments in this framework. In particular, the author claims that the texts under inquiry demonstrate an interconnection between nature and culture that critiques material overconsumption.

I’m not sure if I understand all the rhetorical moves in this article, although the variegated arguments within it are interesting and would be worth reframing. For example, in lines 19-49, we’re introduced to what I thought was going to be the critical thrust of the essay: that is, the materiality of jazz in Beat poetry as emblematic of ecological interconnectedness. As it turns out, jazz and bebop are not the focus at all. Rather, the author transitions to material consumption and anthropocentrism in the next paragraph, which then shapes the argument that follows. 

The author never establishes what the critical stakes of their argument is. They successfully underscore that the poets under inquiry critique extractive capitalism, as lines 507-509 would suggest, and they emphasize this critique on multiple occasions. But I wonder if their close readings (which demonstrate some excellent observations) might interrogate more the how and the why behind Snyder’s and Ginsberg’s polemics. I’m not sure if it is enough, for example, to state that Ginsberg satirizes science or criticizes ecological harm. Rather, I’m curious as to what is the socio-political significance behind the poets’ staging of their critiques within a Buddhist ecological frame? Clarifying the stakes--or the intervention--of the article will certainly help firm up the author’s conclusion, which currently serves as a summary of the essay but does not answer the questions raised in lines 119-121.

All this is to say, I think this essay makes some interesting observations in their close readings that could benefit from more interpretation. I propose resubmission with major revisions to the article's framing and argument. 

Specific Comments

In lines 56-57, the author writes “Evolving from primitive life of prehistoric human beings, human beings began to invent tools and equipment by using the science and technology they mastered.” This is one of the rhetorical moves that I find confusing. If I understand correctly, the author is trying to establish briefly the history of human enterprise, but they foreground this claim concisely on lines 61-62: “In modern and contemporary lives, this anthropocentric cultural activity tends to cause several impacts on human and nonhuman animals as well as on the natural environment.” In fact, that’s all that really needs to be said on the matter given how generalized the previous sentences are. But this is an easy fix. Anthropocentrism and the centering of human life over other species is really the key point in this paragraph. A simple explanation of this term “anthropocentrism” in critical discourse would suffice.

In line 71, the author notes “This animal slaughtering was certainly more or less for material profits.”  The author can be more specific here. It’s worth noting that McClure’s poem was a response to a news report that the US military had machine gunned one hundred whales at the request of the Icelandic government. The article had described the soldiers’ hunt as a heroic enterprise against a dangerous species, which is a reading that McClure’s poem resists. Indeed, his poem is very much a critique on media as well as the military’s attack on nature. 

In lines 88-89, the author notes: “Human overconsumption of material goods will threaten the natural environment. It is because the overconsumption means an overextraction of natural resources as the ingredients of the goods.” This is a general comment, but these lines could be rephrased for clarity. Additionally,  I’m not entirely sure what the author is suggesting in line 113. Does the author mean that material excess reveals the over-valorisation of commodities? What do they mean by objectification in relation to the excerpt of the poem “Ruhr-Gebiet”?

In line 132, the author mentions “biotic materialness.” I’m not sure if this is a common conceptual term, but I’ll point out that there is a huge discourse now on ecological materiality, and it would be worth checking out more recent scholarship in this area. Jane Bennett’s Vibrant Matter comes to mind, for example. I also see Daniel Miller is cited in the bibliography, but not his monograph, Stuff.

I’ll note that in line 153 and 162-3, there is a reference to the author’s emphases, but nothing is bolded or italicized in those sentences. I’m not sure if these are direct quotes.

In lines 191-194, the author makes the claim that the Ginsberg’s and Snyder’s poems reflect a patriarchal materialism that is exemplified by the poets’ use of capitalization. It’s a bit hard to assess the validity of the claim without understanding how the author arrives at this conclusion. The Beats were certainly a male-dominated movement (which Anne Waldman herself has noted!), but I’m not sure how the author connects the materiality of the poem’s language with their observation of patriarchal material cultures.

Further in line 195, the author states: “this poetics at the same time polemicizes ecological issues that become anyone’s responsibility.” What is the move from patriarchal materialism to collective ecological responsibility? Can the article’s author explain more their thought process here?

I’m assuming that the discussion on patriarchal materialism in line 195 was meant to set up the claim in line 257 that the plutonium-related images in Ginsberg’s poem demonstrate masculine authority. It is a claim that I think can be made based on the excerpt the author has provided, although the reference to Whitman is interesting. It makes me wonder if the author should consider how Ginsberg is queering the images of science and extraction in this example (as Ginsberg does in the excerpt provided in lines 315-326).

Lastly, in lines 385-435, I’m also curious about how Orientalism might problematize any reading into Snyder and Ginsberg, since they both embraced Buddhism to a varying extent. Given that ecological interconnection scaffold this article, it would be worth attending to the poets’ appropriation of both Buddhism and American exceptionalism. How do the poets reconcile rebellion with Orientalism, Buddhism, and the quest for Enlightenment?  

Author Response

Thank you very much for your feedback and suggestions.

Yes, I have deleted the word 'bebop' and changed it with 'jazz' since this latter word tends to be suitable with the typography of Ginsberg and Snyder's poems. But indeed, the focus of my analysis is not on jazz poetics but on material poetics. I have read Allen Ginsberg's letters and found that the background why he wrote poems with some ecological overtones is because Ginsberg was in fact an ecology-concerned person. Certainly, his poems also have socio-political overtones, which I have explained briefly the reason why he also depicted the political overtones. Even more so, Gary Snyder as a poet from the West Coast area was a very ecological person. His poems are very nature and environment-based portrayal. "Smokey the Bear Sutra" is one of his poems that clearly raises the environmental issues while it also uses some Buddhist nuances. Both Ginsberg and Snyder use much Buddhist nuances since they had learned Buddhism in a very formal way from Buddhist teachers as well as by living in Japan, India, and Tibet. I have found that their Buddhist poetics purely expresses their features as Beat poets who aimed to criticize mainstream consumerist culture. 

Yes, I have deleted some sentences that I think indeed not necessary to say to streamline the rhetorical moves. I have also clarified my argument about anthropocentric culture, animal hunting as McClure illustrated in his poem.

Yes, I have clarified some sentences about material overconsumption, the relevance between overconsumption and over-extraction of materials, the notion of objectification related to Ginsberg's poem "Ruhr-Gebiet".

Yes, the term 'biotic materialness' is my own coinage. The word 'biotic' here means that any material things are basically derived from living organisms. For instance, fossil fuels that came from plants and animals that have been decayed and sedimented in the earth layers through the ages. But please let me know if the term is indeed not correct to use. Yes, I have added Jane Bennett's Vibrant Matter as one reference. 

I have added my argument about patriarchal materialism and its relevance to capitalization of letters in their poems. I do very apologize if I have not yet explained the answer clearly and missed some argument in responding to your question. Yes, I have added some sentences to clarify the statement that ecological issue is anyone's responsibility.

Yes, I have added some sentences about Ginsberg's poetics as being suggestive of patriarchal materialism. But once again I am very sorry if the explanation is not yet clear. I will be very glad to revise it again later. 

Yes, I have added some sentences about orientalism, American exceptionalism and their relevance to Ginsberg and Snyder's Buddhist poetics. I do very apologize if the argument is not yet convincing enough. Thank you very much.

Reviewer 3 Report

The essay shows substantial research in secondary sources. It reviews the field extensively. Its argument hardly breaks any new ground in thinking about the Buddhist — Ecological inflections of these Snyder and Ginsberg poems, much less how these play out more broadly in Beat poetry and thinking.

The Abstract demonstrates the deficiencies of the argument and the sometimes confusing syntax and expression of the essay as a whole. And the essay, I think, tries to do too much, and claim too much for Plutonian Ode and Smokey the Bear Sutra and it struggles to make a convincing case for why these two particular poems might especially exemplify the ecological concerns of both poets. In its introduction, the essay also attempts to argue that the influence of jazz — especially Bop — on the Beats is a means for Beat poetics to offer a radical ecological critique of anthropocentrism. This claim is unproven (and the specific poems it then analyses are from periods in the poets’ writing significantly after bop was a radical force in American culture). 

The close readings of each poem — though hugely informed by secondary material and research — are also rather ‘literal’ in their readings of the poems — glossing references and remarking on patterns of emphases provided by capitalisations of noun phrases. The do not sufficiently push the analyses to support and develop the argument about how these poems (and Beat poetics more broadly) work through Buddhism in order to press for an ecological argument. 

There are a number of grammatical and other writing errors (misused and missed indefinite articles especially) that blunt the precision and force of the argument.

Author Response

Thank you very much for your feedback and suggestions.

Yes, I have revised the abstract by deleting and changing some words and sentences to reduce wordiness, to tighten and streamline it. Yes, I have changed the word 'bebop' with 'jazz' to clarify that the influence of jazz in general on Ginsberg and Snyder's poems is still significant. Indeed, bebop was popular in America in the late 1940s and early 1950s but jazz tradition has kept going since then. The long lines and enjambement of their poems for instance exemplify improvisation of jazz choruses. Ginsberg for instance was inspired by Lester Young as Kerouac by Charlie Parker and Thelonious Monk. While Snyder tends to be inspired by jazz musicians in general.

So far, I have always tried to read Beat poetry from ecopoetic perspectives. In fact, Ginsberg's "Plutonian Ode" and Snyder's "The Smokey Bear Sutra" are some of Beat poems that clearly contain issues about environment. In addition, both Ginsberg and Snyder were affiliated with Buddhism. They not only practiced some Buddhist ways such as doing sitting (meditation), chanting mantra, but also were fascinated with Buddhist teachings that they deploy in many of their poems. I think the long lines of Ginsberg and Snyder's poems also signify Buddhist mantra in some ways, while the polemics they raise serve as the messages the mantra convey to each individual being. The interconnected lines in their poems also represent the ecology of the natural environment.

Yes, I have corrected some grammatical errors including some missing indefinite articles. But I do very apologize if I have missed revising some parts.

Thank you very much.

Round 2

Reviewer 2 Report

Unfortunately, the structural issues in the essay have not been addressed. While the author has clearly made some revisions, the edit is not sufficient to fix the underlying weaknesses of the argument. I was struggling to follow the trajectory of this paper given that the key conceptual framework ("biotic materialness") isn't clearly explained. Nor has the author foregrounded that this term serves as their critical intervention. 

The essay also betrays substantial grammar, formatting, and spelling errors and would require reediting to enhance the argument's clarity.

If the author wishes to revise this paper fully, I suggest they narrow their focus and readings. Their close readings remain surface-level and observational and any revision should really expand on interpretation. 

Author Response

Dear Reviewer,

Thank you very much for your review, feedback and suggestions. I've revised my title from 'biotic materialness' to 'anthropocentric materialism'. The reason why I changed this term is because I would like to focus my analysis on human overuse of material things as Ginberg describes through 'plutonium' and Snyder portrays in material things humans use. The former term 'biotic materialness' is too broad and only refers to 'materiality' of things. Meanwhile, what I would like to argue in my article is the polemics the poets raise in their poems indicate human material engrossment, which means materialism.

So far I have tried to shorten wordiness of my sentences and checked the grammar, format, spelling of my writing. I do very apologize if I have still missed several parts. 

As I have told that by changing the title into 'anthropocentric materialism', I am trying to focus my analysis on human overuse of material things as materialism. But my analysis of this materialism also consults with Buddhist principles of insubstantiality of phenomena in an ultimate sense as the poets suggest in their poems.

Reviewer 3 Report

The recent corrections, edits and clarifications to the essay have certainly improved its clarity and ironed out some of the conceptual difficulties and leaps that were initially present. And I appreciate the attention that the author has given to the initial report on the essay.

I still find the essay very densely packed with far too many concepts and strands in its argument. That Ginsberg and Snyder are ‘jazz’ poets, seems largely irrelevant to the central argument about biotic commodities and the raising of ecological awareness in the two poems discussed. Jazz / beat / bebop poetics does not — necessarily — inhere in long, enjambed lines (see, for example, Whitman — probably a better model than jazz for this particular AG poem). And the relationship of jazz to ecological thinking is nowhere established in the essay.

That both poets are of ‘Beat’ descent, as with jazz, seems also largely irrelevant, too. All that needs establishing — rather than a long and detailed genealogy of each poet’s relationship to the Beats, and of how Beat poetics is ecological — is the countercultural nature of their critique of American consumerist and materialist values.

The essay goes to great lengths in describing various aspects of AG and GS’s Buddhism (though it makes no mention of GS’s influential essay ‘Buddhism and the Coming Revolution’). Again, all of this seems rather unnecessary — as those aspects of it that are relevant to the critiques made by their respective poems will be (should be) made apparent and discussed as part of the analysis of each poem.

So, although the first half of the essay establishes the vast amount of background research that has gone into the essay, the very nature of that vast amount of research material gets in the way of the essay’s clarity of purpose. It is trying to say far too much; and this dulls the impact of its essential argument.

This much is apparent — still — in the Abstract and how the argument is set out in the introduction. Though much improved and tightened, the two research questions still feel somewhat unfocused. The array of terms and concepts introduced — biotic commodities, Buddhism, jazz, raising of ecological awareness, beat poetics, American consumerism, materialism, etc — overload the essay, and mean that it cannot cut a clean path through all of them. Though this abundance is testimony to the author’s dedication, enthusiasm and sheer hard work in researching and attempting to synthesise all these sources, it leaves the actual discussion and analysis of the two poems, ‘Plutonian Ode’ and ‘Smokey the Bear Sutra’ to rather skate on the surface of all this abundance.

The analyses of the poems, then, are rather superficial and pedestrian — working their way through, line by line, or strophe by strophe, each poem. They read as commentaries rather than critiques of the poems. This is because the essay hasn’t really established what it is arguing about the poems, and thus it doesn’t quite know what to do with its reading of them, other than gloss them. Indeed, it seems to completely miss the wry and ironic humour of Snyder’s poem — turning a Forest Service publicity campaign slogan about putting out cigarette butts into a sutra to contemplate ecological destruction is surely using laughter / humour as an instructional tool (vey much in line with GS’s own Buddhist temperament). And the tone in GS’s poem is a polar opposite of that in AG’s. His poem is angry, portentous and adopts the prophetic and bardic tone typical of him in this sort of polemic. Much more detail is needed on how these two poems differ; this means far more attention needs giving to the poetic nuances of each poem.

So, in the end, I am still not convinced by the essay. I think it feels very much like it struggles to see (to adopt an ecological metaphor) the wood for the trees. 

If the author still wants to pursue publication, then the essay needs a really radical overhauling. It should START with the close readings of the two poems — really push them for what they discover of AG’s and GS’s pressing of their respective Buddhist thinking and their poetics towards the raising of ecological consciousness in their readers. Doing this would then help the author recognise what of the copious other, supporting, material in the essay is strictly necessary to the argument being made. Anything not strictly necessary should be ditched. The argument needs a clear path through the forest.

Author Response

Dear Reviewer,

Thank you very much Sir for the rigorous feedback and suggestions.  Yes, I mentioned jazz influences in Ginsberg and Snyder's poems but indeed I do not focus the discussion on this jazz influence in my article. Frankly speaking this idea of jazz influence was based on my argument in my dissertation, in which I discussed spontaneity in Beat poetry, while the Beat poets adopted this spontaneity from bebop jazz and from a spontaneous way of behaving as Mahayana Buddhism has evoked due to the notion of insubstantiality of external phenomena. My former argument about the relevance between spontaneity in jazz and that in Buddhism is that both spontaneities mean to turn away from materialism. In bebop, the spontaneity means to revolutionize commercialization of big band (swing jazz) of the earlier era (for this argument, I did some research about the nature of bebop from some books). In Mahayana Buddhism, this spontaneity refers to one'a non-attached behavior to material things since one has an insight into the true nature of phenomena (this argument is also based on books on Buddhism that I have read). But again, I do not particularly want to talk about bebop spontaneity and its influence on Beat writings in this article.  

I've added Snyder's essay "Buddhism and the Coming Revolution" as you have suggested. I have tried to read again my article and reduced the wordy parts. But I do very apologize if this latest revision still has many shortcomings as you have expected me to change. 

I do very apologize if I do not really undertand what you mean that my analysis is just a commentary rather than a critique of the poems. This is the best I can do for now. If my analysis should be a critique rather than just a commentary, I think I will need more time and more thoughts to change the way I read and analyze the poem as you have asked. While at this present, I have some other writing commitments to do, so that it might not be possible to think through a different or innovative interpretation of the poems. So back to the title of my article, I just would like to discuss material poetics in their poems as their criticism of human materialism. Since the special issue of this Humanities journal is about Beat eco-reading, I also argue that Ginsberg and Snyder's material poetics and their criticism in the poems certainly aim to raise anyone's ecological awareness. For Allen Ginsberg's poetics, I have consulted with some books about his poetics. One of the books, for instance, was written by a Beat scholar and a social activist who was once a student and best friend of Allen Ginsberg. In one chapter about "Plutonian Ode", he also discussed that Ginsberg's plutonian poetics means to raise public's awareness of the hazard of the material. I also consulted with Ginsberg's biography, in which he described the event when he and some of his friends blocked the Rocky Flats railroad tracks to halt "trainload of waste fissile materials" after he completed writing the poem "Plutonian Ode". 

I do very apologize if I have not yet elaborated the differences between Ginsberg and Synder's poems. Of course, everybody can see if in terms of form and content, their poems are different. But in my analysis, again I focus on their material poetics that serves as criticism of materialism and as vision to evoke one's ecological awareness. In this case, I have referred to Buddhist principles in interpreting and analyzing their material poetics as being suggestive of ecological interdependence and being devoid of true nature. This vision means to enlighten anyone not to overconsume any material thing, which means that this behavior will conserve the biotic life and any life form. 

Thank you very much Sir.

Round 3

Reviewer 3 Report

The essay is leaner and the recent changes do tighten and clarify the argument about ecological consciousness and how this is raised by each poem. 

Though I still think the essay could be better focused and clarified, I think it does now warrant publication; but it needs a close proofreading. Some slips have crept in where material has been deleted or inserted. And any slips in the use of English need to be caught by copy editors. 

Author Response

Dear Reviewer,

Thank you very much for your latest feedback and suggestions. Yes, I have made minor revisions to the recent draft of my article. I've added some sentences to clarify my argument. 

Thank you very much once again. Wishing you have a good day.

Best regards,

Henrikus